# Spherical Polyelectrolyte Brushes as Templates to Prepare Hollow Silica Spheres Encapsulating Metal Nanoparticles

**DOI:** 10.3390/nano10040799

**Published:** 2020-04-21

**Authors:** Qingsong Yang, Li Li, Fang Zhao, Yunwei Wang, Zhishuang Ye, Chen Hua, Zhiyong Liu, Klemen Bohinc, Xuhong Guo

**Affiliations:** 1State Key Laboratory of Chemical Engineering, School of Chemical Engineering, East China University of Science and Technology, Shanghai 200237, China; yangqingsong@mail.ecust.edu.cn (Q.Y.); fzhao1@ecust.edu.cn (F.Z.); yunweiwang@mail.ecust.edu.cn (Y.W.); zhishuangye@mail.ecust.edu.cn (Z.Y.); huachecust@gmail.com (C.H.); 2Engineering Research Center of Materials Chemical Engineering of Xinjiang Bingtuan, Shihezi University, Shihezi 832000, Xinjiang, China; lzy_tea@shzu.edu.cn; 3Faculty of Health Sciences, University of Ljubljana, 1000 Ljubljana, Slovenia; bohinck@zf.uni-lj.si

**Keywords:** spherical polyelectrolyte brush, hollow silica, silver nanoparticles, catalyst

## Abstract

Integrating hollow silica spheres with metal nanoparticles to fabricate multifunctional hybrid materials has attracted increasing attention in catalysis, detection, and drug delivery. Here, we report a simple and general method to prepare hollow silica spheres encapsulating silver nanoparticles (Ag@SiO_2_) based on spherical polyelectrolyte brushes (SPB), which consist of a polystyrene core and densely grafted poly (acrylic acid) (PAA) chains. SPB were firstly used as nanoreactors to generate silver nanoparticles in situ and then used as sacrificial templates to prepare hybrid hollow silica spheres. The resulted Ag@SiO_2_ composites exhibit high catalytic activity and good reusability for the reduction of 4-nitrophenol to 4-aminophenol by NaBH_4_. More importantly, this developed approach can be extended to the encapsulation of other metal nanoparticles such as gold nanoparticles into the hollow silica spheres. This work demonstrates that SPB are promising candidates for the preparation of hollow spheres with encapsulated metal nanoparticles and the resulted hybrid spheres show great potential applications in catalysis.

## 1. Introduction

Hollow silica spheres which consist of a porous shell and large central cavity have attracted wide attention in the fields of drug delivery [1], catalysis [2,3], detection [4], and antibacterial area [5] due to their unique properties like low density, large specific surface area, and easy functionalization [6,7,8,9]. To fully exploit the hollow structures, functional nanoparticles can be further encapsulated inside the interior cavities as well as the pores and channels of the shell, resulting in highly versatile composites [10,11]. A great variety of active species have been combined with hollow silica spheres such as metal nanoparticles [12], metal oxides [13,14], and carbon quantum dots [15] to endow the hybrid spheres with extra catalytic, magnetic or photoluminescence properties.

In the field of catalysis, multiple metal or metal oxides nanoparticles that are encapsulated within hollow silica spheres have received particular interests due to the following advantages [16,17]. Firstly, the porous silica shell can provide better protection against agglomeration or poisoning of the catalysts inside the cavity, and better control for specific diffusion of reactants and products between the interior and outer environment. Secondly, the small catalysts encapsulated by silica spheres would be much easier separated from the heterogeneous reaction system by centrifugation or suction filtration for reuse [14,18].

To date, several methods have been developed to construct hybrid silica nanoparticles based on different mechanisms and physical or chemical processes including the template method [2,19,20], Ostwald ripening process [21], spray techniques [22], and microfluidics techniques [23], among others [10]. Among them, sacrificial template is probably the most widely investigated method due to the well-controlled size and monodispersed morphology of the prepared particles. For example, Ni et al. used polystyrene microsphere as sacrificial templates to fabricate silica hollow microspheres with noble metal (Au, Pt, and Pd) nanoparticles inside their cavities [19]. Although great achievements have been made by these methods, there are still some challenges in the synthesis and practical applications of hybrid hollow silica nanoparticles. Typically, the strategy of these reported methods is firstly fabricating the encapsulated inorganic nanoparticles and then coating these nanoparticles on the surface of the templates. Normally, the controlled synthesis of small functional nanoparticles and depositing of these nanoparticles onto the surface of templates are both challenging. Thus, it is of great importance to simplify the multi-step synthetic processes, especially avoiding the complicated operation for the conjunction of nanoparticles and templates.

Recently, Li et al. prepared hollow mesoporous silica nanoreactors with small-size metal oxide nanoparticles by deposition of silica onto metal-containing charge-driven polymer micelles and subsequent calcination [16,24]. This method simultaneously forms hollow silica spheres and metal oxide nanoparticles in cavities, which is general for the preparation of hollow silica nanoparticles with metal oxide nanoparticles. However, the use of supramolecular polyelectrolyte may limit its industrial applications. Therefore, a simple and general method to prepare hybrid hollow silica nanospheres encapsulating functional nanoparticles is still highly desired.

Spherical polyelectrolyte brushes (SPB), which consist of a spherical core and densely grafted polyelectrolyte chains, have drawn much attention in catalysis [25], protein separation [26,27], and detection [28,29]. Previous works have demonstrated that SPB can be used as nanoreactors for in situ generation of metal nanoparticles due to their excellent ability of confining counterions. Various inorganic nanoparticles such as Pt [30], Ag [31], Ni [32], and MnO_2_ [33] have been prepared by SPB, which showed excellent stability and catalytic performance. Although SPB were demonstrated to be ideal nanoreactors for the preparation of these small nanoparticles, the separation of SPB with immobilized nanoparticles from aqueous reaction system for reuse is still challenging because of the low-density difference of SPB with water, which highly limits their practical applications. However, our recent works demonstrated that SPB were also ideal templates to fabricate hollow silica spheres with controllable structures [34,35,36]. Logically, we hypothesize that the abilities of generating and immobilizing metal nanoparticles as well as fabricating hollow silica spheres by SPB may be combined to create a unique platform for the preparation of hollow silica spheres encapsulating functional nanoparticles. The resulted hybrid nanoparticles are expected to possess more complex structures and more attractive properties.

In this work, we report a facile and general strategy to prepare hollow silica spheres encapsulating metal nanoparticles based on SPB. Firstly, using SPB as nanoreactors to generate metal nanoparticles in situ and then using the metal nanoparticles loaded SPB as templates to prepare silica shell. Finally, removing SPB by calcination to obtain the hollow silica spheres decorated with metal nanoparticles inside their cavities and porous shells. Hollow silica spheres encapsulating silver nanoparticles, designated as Ag@SiO_2_, were successfully produced by this method. The catalytic performance and reusability of Ag@SiO_2_ were investigated by using the reduction of 4-nitrophenol as the model reaction. In addition, gold nanoparticles were also encapsulated into the hollow silica spheres by the similar method, which demonstrates that this method possesses versatility in encapsulating metal nanoparticles. Compared with bare metal nanoparticles, the hybrid hollow spheres are more easily collected by certification and show wider potential applications in catalysis, detection, and antibacterial area.

## 2. Materials and Methods

### 2.1. Materials

Styrene and acrylic acid (AA) were purchased from Sinopharm Chemical Reagent Co., Ltd. (Shanghai, China) and purified by vacuum distillation. Potassium persulfate (KPS), sodium dodecyl sulfonate (SDS) and methacryloyl chloride (MC) were purchased from Aladdin Reagent Co., Ltd. (Shanghai, China). Tetraethyl orthosilicate (TEOS), 4-nitrophenol (4-NP), sodium borohydride (NaBH_4_), hydrogen tetrachloroaurate (III) tetrahydrate (HAuCl_4_·4H_2_O), and silver nitrate (AgNO_3_) were purchased from Shanghai Lingfeng Chemical Reagent Co., Ltd. (Shanghai, China). 2-Hydroxy-40-hydroxyethoxy-2-methyl propiophenone (HMP) was obtained from Tokyo Chemical Industry Co., Ltd. (Tokyo, Japan). Absolute ethanol (C_2_H_5_OH, 99.7%) and ammonia solution (NH_3_·H_2_O, 25 wt%) were purchased from Shanghai Titan Scientific Co., Ltd. (Shanghai, China). Ultrapure water was obtained by reverse osmosis (Millipore Milli-Q) and used throughout the experiments. The photoinitiator 2-[p-(2-hydroxy-2-methylpropiophenone)]-ethylene glycol methacrylate (HMEM) was synthesized in our laboratory according to the literature [37].

### 2.2. Preparation of SPB

The polystyrene (PS) core latexes were synthesized by a conventional emulsion polymerization. In brief, 0.16 g SDS, 0.48 g KPS, 8.0 g styrene, and 200 mL ultrapure water were added into a three-necked round-bottom flask. The system was purged by repeated degassing and subsequent flushing with nitrogen for five times to remove oxygen. The polymerization was carried out at 80 °C at a stirring rate of 300 rpm·min^−1^. After 2 h, the reaction temperature was reduced to 70 °C. Then, 0.8 g HMEM dissolved in 7.2 g acetone was added into the reaction system under a “starved” condition (6 s per drop). After another 1 h, the PS core coated with a thin shell of photoinitiator HMEM was obtained and then purified by dialysis using ultrapure water until the conductivity of outer water was kept constant.

The SPB with a PS core and poly (acrylic acid) (PAA) chains were prepared by photo emulsion polymerization. Typically, 200 g PS core latex (0.7 wt%) and 1.4 g monomer AA were added into a homemade quartz reactor. The reactor was then purged by repeated evacuation and subsequent addition of nitrogen. Photo polymerization was carried out under UV radiation with vigorous magnetic stirring. After reaction for 2 h, the SPB latex was collected and purified by ultrafiltration using ultrapure water until the conductivity of outer water was kept constant.

### 2.3. Preparation of Hollow Ag@SiO_2_ and Au@SiO_2_

Using the prepared SPB as nanoreactors, silver and gold nanoparticles were generated by the reduction of corresponding metal ions (Ag^+^ and AuCl_4_^−^) with NaBH_4_. In a typical run, 5.0 g SPB latex (2.0 wt%, pH = 7.0) was dispersed into 90.0 g ethanol solution under mechanical stirring conditions (300 rpm·min^−1^). Then, 0.0170 g AgNO_3_ (0.0723 g HAuCl_4_·4H_2_O) dissolved in 2.5 mL pure water was added into the solution slowly and the mixture was stirred for 2 h at room temperature. Next, a freshly prepared aqueous solution of NaBH_4_ was injected into the above solution rapidly. The mixture was stirred for another 1 h after the solution turned into light yellow (wine red).

Hollow Ag@SiO_2_ (Au@SiO_2_) composite nanoparticles were obtained by using the silver (gold) loaded SPB as templates. About 2.0 g ammonia (25 wt%), which acts as a base catalyst, was added to the above reaction system to adjust its pH to 11.0. After stirring for 10 min, 0. 75 g TEOS which acts as silica precursor dissolved in 5 mL ethanol was slowly added into the above solution at a rate of 2 mL·h^−1^. After the addition of TEOS, the reaction lasted for 21 h at room temperature to complete the crystallization process. Then, the SPB@Ag@SiO_2_ (SPB@Au@SiO_2_) composites were obtained by repeated centrifugation, washing with ethanol and ultrapure water three times, respectively. Finally, the yellow (red) solids were collected and dried in an oven at 40 °C overnight and further calcined at 550 °C for 3 h in the air to obtain hollow Ag@SiO_2_ (Au@SiO_2_) nanoparticles.

### 2.4. Catalytic Reactions with Hollow Ag@SiO_2_

The catalytic performance of Ag@SiO_2_ nanocomposites was investigated using the reduction of 4-NP by NaBH_4_ as the model reaction. Typically, 1.5 mL of 4-NP aqueous solution (0.2 mM), and 1.5 mL of freshly prepared NaBH_4_ solution (40 mM) were added into a quartz cuvette. Then, 0.1 mL of aqueous solution with certain concentrations of hollow Ag@SiO_2_ spheres was added into the mixture to catalyst the reaction. The absorbance at 400 nm was recorded by UV-vis spectroscopy as a function of time. The color of the solution changed from yellow to colorless gradually with the reaction proceeding. To investigate the reusability of the prepared catalyst, the reduction reaction was amplified ten times. After the reaction, the catalyst was collected by centrifugation and washed with water twice, and reused for the next run, which was repeated for eight times.

### 2.5. Characterization

Transmission Electron Microscopy (TEM) characterization was performed using the JEM-1400 and JEM-2100 electron microscopes (JEOL, Tokyo, Japan). Thermogravimetric analysis (TGA) was performed on a Perkin-Elmer Pyris 1 TGA instrument at a heating rate of 20 °C min^−1^ from 50 to 800 °C in air atmosphere (PerkinElmer, Waltham, Ma, USA). The UV-vis absorption spectra were recorded by UV-vis spectrophotometer (UV-2550, shimadzu, Kyoto, Japan). X-ray diffraction (XRD) was performed by Bruker D8 Advance X-ray diffractometer with Cu Kα radiation in the 2θ range from 10° to 90° (Bruker, Karlsruhe, Germany). The actual metal loadings content was determined by using an Agilent 725 inductively coupled plasma optical emission spectrometer (ICP-OES) (Agilent, Santa Clara, CA, USA). Brunauer-Emmett-Teller (BET) surface area, pore size distribution as well as the adsorption and desorption isotherms of Ag@SiO_2_ were determined by N_2_ adsorption at 77 K, using a Micromeritics ASAP-2020HD88 automatic specific surface area and porous physical adsorption analyzer (Micromeritics, Norcross, GA, USA).

## 3. Results

### 3.1. Preparation and Characterization of Ag@SiO_2_

The preparation strategy of hollow silica spheres containing silver nanoparticles is shown in Figure 1, which can be mainly divided into three stages: preparation of SPB, in situ generation of silver nanoparticles, and fabrication of hybrid hollow silica spheres. In the first stage, the SPB with a PS core and PAA chains were prepared by photo emulsion polymerization and characterized by TEM and DLS. Figure 2a,b show the TEM images of synthesized PS and SPB, respectively, indicating that both of them are highly monodispersed with spherical morphology and smooth surface. The DLS results, as shown in Appendix A, demonstrate that PS core and SPB display a diameter of approximately 110 nm and 449 nm with a polydispersity value of 0.002 and 0.025, respectively, which means that the PS core and SPB have narrow size distribution and the thickness of PAA brush layer is about 169.5 nm.

In the second stage, the prepared SPB were used as nanoreactors to generate silver nanoparticles in situ. As known, the dissociated carboxylic acid groups in PAA chains can provide negative charges to confine counter ions. Thus, when AgNO_3_ solution was added into the reaction system, most of the silver ions would be captured by PAA chains through electrostatic interactions [38]. With the addition of NaBH_4_ as the reducer, the nucleation and growth of silver nanoparticles would mainly happen around the PAA chains, resulting in SPB with immobilized silver nanoparticles (SPB@Ag). The generation of silver nanoparticles was observed from the change in color of the solution from milk white to light yellow. After the reduction of Ag^+^ by NaBH_4_, some carboxyl groups should be released and be able to capture counter ions again [39].

In the third stage, the SPB@Ag composite nanoparticles were further used as templates to prepare silica shell by a modified Stӧber method [34,35]. The PAA shell of SPB also plays a vital role in the generation of silica shell. Firstly, the carboxylic groups of PAA shell can effectively confine the ammonium ions, which act as catalysts for the sol-gel reaction. Secondly, the PAA shell as one of the excellent water-absorbent materials can absorb and retain the water, which is one of the reactants of the hydrolysis reaction of silica precursors [40]. Thus, when the silica precursor (TEOS) was added into the reaction system, the hydrolysis reaction would be mainly carried out in PAA shell, resulting in a large amount of silica nanoparticles. With the growth and aggregation of these silica nanoparticles, the silica shell was generated. After calcination of the obtained SPB@Ag@SiO_2_ composite nanoparticles in air at 550 °C for 3 h, the hollow silica spheres with encapsulated silver nanoparticles (Ag@SiO_2_) were prepared.

Figure 2c,d display the TEM images of hollow Ag@SiO_2_. It is clear that the Ag@SiO_2_ spheres possess a well-defined spherical shell and cavity structures, with a uniform cavity diameter and shell thickness of about 103 nm and 20 nm, respectively. The cavity diameter of Ag@SiO_2_ is close to the diameter of PS core, which implies that the silica shell is formed around the PS core. Some small dark particles can be observed on the cavity surface and silica shell, indicating the presence of silver nanoparticles. Since the silver nanoparticles were generated within the PAA brush layer and so did the silica shell, it is reasonable that some silver nanoparticles were embedded inside the silica shell. The average size of encapsulated silver nanoparticles is about 3.4 nm (size distribution is shown in Appendix A), which is smaller than that reported in some papers [17,41,42]. The amount of silver content in hollow silica spheres was determined to be 2.8 wt% by ICP-OES.

To validate the formation and distribution of Ag nanoparticles, dark-field image of hollow Ag@SiO_2_ and elemental mapping of Ag@SiO_2_ nanospheres were also employed, as shown in Figure 3. It clearly shows the core shell structure of Ag@SiO_2_ nanoparticle with some nanoparticles in the cavity surface and silica shell. From the EDX elemental mapping of Si, O, and Ag in the hollow Ag@SiO_2_ spheres, we can observe that the profiles of Si and O elements are mainly located in silica shell and the silver elements are uniformly distributed in silica shell and the inner surface of the hollow sphere.

The obtained Ag@SiO_2_ nanospheres were further characterized by UV-vis, TG analysis, and BET. The UV-vis spectrum of hollow Ag@SiO_2_ is shown in Figure 4a. An obvious absorption peak at about 407 nm is visible, which confirms the generation of silver nanoparticles as the typical adsorption peak of silver nanoparticles is around 400 nm. The TG analysis results of SPB@Ag@SiO_2_ and hollow Ag@SiO_2_ spheres are displayed in Figure 4b, which confirm the removal of SPB and generation of hollow silica spheres. The weight loss of SPB@Ag@SiO_2_ was about 33%, while almost no weight loss was observed of hollow Ag@SiO_2_, which suggests that the polymer compounds and organic components of silica shell were completely decomposed at the calcination temperature of 550 °C. Figure 4c shows the N_2_ adsorption and desorption isotherm of hollow Ag@SiO_2_ nanospheres, which can be classified as a type-IV curve. A hysteresis loop was observed at higher N_2_ pressure (*P/P_0_* = 0.45–0.95), indicating the coexistence of mesopores [43]. The measured BET surface area and volume of pores are 348.80 m^2^ g^−1^ and 0.53 cm^3^ g^−1^, respectively. The pore size distribution calculated by BJH method shows a strong peak centered at 4.0 nm, as shown in Figure 4b, which might be the size of mesopores in the silica shell causing by the removing of PAA chains. As the nucleation and growth of silica nanoparticles happened along the PAA chains, the PAA chains would be embedded in the resulted silica shell. As a result, when the organic components were removed by calcination, some small pores were formed. The porous silica shell makes the hollow Ag@SiO_2_ quite suitable for rapid mass transport, which is beneficial for the catalytic applications.

These characterization results confirm that silver nanoparticles were successfully generated and mainly located in the silica shell. Thus, using SPB as both nanoreactors and templates is an effective method for encapsulation silver nanoparticles inside hollow silica spheres. The encapsulated silver nanoparticles with very small average size (3.4 nm) should possess high catalytic activity.

### 3.2. Catalytic Performance of Ag@SiO_2_

The catalytic performance of prepared Ag@SiO_2_ spheres was investigated using the reduction of 4-nitrophenol (4-NP) to 4-aminophenol (4-AP) by NaBH_4_ as the model reaction, which is an important chemical conversion reaction from both industrial and environmental perspectives [44]. The reaction process can be easily monitored by UV-vis spectroscopy as the 4-NP ions exhibit a typical absorption peak at 400 nm [45]. Figure 5a shows the typical UV-vis spectra for the reduction of 4-NP using Ag@SiO_2_ nanospheres as the catalyst. When hollow Ag@SiO_2_ nanoparticles were added into the reaction system, the absorption peak at 400 nm decreased gradually with time, indicating the reduction of 4-NP. Four isosbestic points at 225, 245, 279, and 314 nm were observed, implying the dominant product of 4-AP in the reaction [31]. The reaction process was also monitored under time dependence pattern as shown in Figure 5b, an induction time for the reaction was observed, which is a normal phenomenon of heterogeneous catalysis because of the activation of catalysis in the reaction mixtures [32]. In order to prove the catalytic activity of Ag@SiO_2_ spheres originating from the encapsulated silver nanoparticles, the reaction was carried out under the same conditions, except when using pure silica spheres as the catalyst. Almost no change in the absorption intensity at 400 nm was observed, indicating that the pure silica spheres did not possess catalytic activity.

Since the amount of NaBH_4_ is in great excess in our experiments (*C_(NaBH4)_/C_(4-NP)_* = 200:1), the reaction can be considered as a pseudo-first order reaction. The apparent kinetic rate constant *K_app_* should be proportional to the total surface *S* of all metal nanoparticles as the catalysis takes place on the surface of catalyst. Thus, the *K_app_* and kinetic constant *K_1_* in this reaction system can be calculated according to Equation (1) [45,46]:(1)$$\frac{dC_{t}}{dt}=-K_{app}C_{t}=K_{1}SC_{t}$$
where *C_t_* represents the concentration of 4-NP at time *t*, and was calculated from the relative intensity ratio of the respective absorbances at 400 nm. The surface area *S* was calculated by assuming that the silver nanoparticles were spherical homogeneous spheres with diameter of 3.4 nm and the bulk density of silver nanoparticles was 10,500 kg m^−3^ [47].

The time dependence of the adsorption intensity at 400 nm of the reaction system with different concentrations of Ag@SiO_2_ was investigated. The absorption intensity decreased with reaction time and the decrease rate increased with higher concentration of Ag@SiO_2_ spheres. By fitting ln (*C_t_/C_0_*) as a function of the reaction time *t*, linear relationships were obtained as shown in Figure 6a, indicating that the reaction matches the first-order reaction very well. Thus, the apparent rate constant *K_app_* can be determined from the slope of fitting curve and *K_1_* was further calculated according to Equation (1). When the silver concentration was 0.27 mg·L^−1^, the *K_app_* and *K_1_* were calculated to be 0.0050 s^−1^ and 0.22 L s^−1^ m^−2^, respectively, which reflect the high catalytic activity of Ag@SiO_2_ spheres for the reduction reaction of 4-NP to 4-AP as the value of *K_app_* in most of papers is around 10^−3^ s^−1^ [31,32,48,49]. The reaction rate constant *K_app_* is linear with the concentration of the Ag@SiO_2_ as shown in Figure 6b, which is consistent with the previous report that the reaction rate constant is proportional to the catalyst concentration for heterogeneous or microheterogeneous catalysis [32,46].

Similarly, the liner relations between ln(*C_t_/C_0_*) and reaction time *t* at different temperatures (from 283.15 K to 303.15 K) were obtained, as shown in Figure 6c. The corresponding values of *K_app_* were calculated and the plot of ln (*K_app_*) versus *1/T* is displayed in Figure 6d. The apparent activation energy (*E_a_*), which reflects the temperature dependency of the reaction rate constant, was calculated as 81.80 KJ mol^−1^ from the slope of fitting curve in Figure 6d. The obtained *E_a_* is higher than that of silver nanoparticles immobilized in SPB, where *E_a_* was reported to be 60.27 KJ·mol^−1^ [31], which is probably because the value of *E_a_* is relevant to the structure of catalyst and carrier.

The reusability of hollow Ag@SiO_2_ spheres was also investigated since it is a crucial concern for the practical application of catalysts. The Ag@SiO_2_ spheres can be much easier separated from an aqueous dispersion by centrifugation compared with the unencapsulated silver nanoparticles or SPB@Ag. The collected Ag@SiO_2_ spheres were reused in the next round of the reaction. The reaction conversion at reaction time *t* was calculated according to Equation (2):(2)$$Conversion\left(\%\right)=\left(1-\frac{C_{t}}{C_{0}}\right)\times 100$$

As shown in Figure 7, the conversion of each cycle run maintains above 95% for a reaction time of 40 min, which means that the catalytic activity of Ag@SiO_2_ was maintained at least until the eighth use. The good reusability of the Ag@SiO_2_ spheres is probably attributed to the effective protection of silica shell. These results confirm that the hollow Ag@SiO_2_ is an excellent reusable catalyst with high conversion, which would be favorable for its practical applications.

### 3.3. Synthesis of Hollow Au@SiO_2_ Nanoparticles

Since SPB has been successfully used as a nanoreactor to prepare a series of metal nanoparticles, our strategy should be applicable to the encapsulation of these nanoparticles into hollow silica spheres. Thus, hollow silica spheres encapsulating gold nanoparticles (Au@SiO_2_) were also prepared using the similar method as described above. The auric chloride ions can be confined by SPB because of the strong coordination interaction between the metal moieties of gold precursors and the carboxyl groups of PAA chains [50]. The gold nanoparticles were then generated in PAA chains with the addition of NaBH_4_ solution.

As shown in Figure 8a, hollow silica spheres with gold nanoparticles inside the cavities and silica shell can be clearly observed. The gold nanoparticles have an average diameter of 6.6 nm. The presence of gold nanoparticles was further confirmed by the UV-vis spectrum due to the existence of the typical absorption peak of gold nanoparticles at 529 nm, as shown in Figure 8c. The structure of the hollow Au@SiO_2_ nanospheres was investigated by X-ray powder diffraction (XRD). From the patterns of hollow Au@SiO_2_ nanoparticles, a broad peak at 2θ = 22.27° can be observed, which indicates the presence of amorphous silica. In addition, four appreciable diffraction peaks located at 2θ = 38.15°, 44.46°, 64.66°, and 77.55° were detected, which can be assigned to (111), (200), (220), and (311) reflections of the gold lattice, respectively [51]. The amount of gold content in hollow silica spheres determined by ICP-OES is about 7.0 wt%. The successful encapsulation of gold nanoparticles demonstrates that our preparation strategy based on SPB is highly versatile for the encapsulation of metal nanoparticles inside cavity and the porous shell of hollow silica spheres.

## 4. Conclusions

In summary, we report a facile and general method to prepare hollow silica spheres encapsulating silver nanoparticles (Ag@SiO_2_) based on SPB. A key feature of the synthetic strategy is that the SPB with a PS core and PAA chains work as both nanoreactors for the generation of silver nanoparticles and templates for the fabrication of hollow silica spheres. The prepared hollow Ag@SiO_2_ composite nanospheres, which can be easily separated from the reaction solution by centrifugation, exhibit high catalytic activity and good reusability for the reduction of 4-NP to 4-AP by NaBH_4_. More importantly, this preparation method can be applied to the encapsulation of other metal nanoparticles such as gold nanoparticles into the hollow silica spheres, resulting in multifunctional nanocomposites. This work demonstrates that in situ generation of metal nanoparticles and silica shell based on SPB is an effective strategy for encapsulating metal nanoparticles into the silica spheres. The unique structure and properties of hollow hybrid nanospheres will make them ideal candidates for potential applications in catalysis, sensing, and detection.

## Figures and Tables

**Figure 1 nanomaterials-10-00799-f001:**
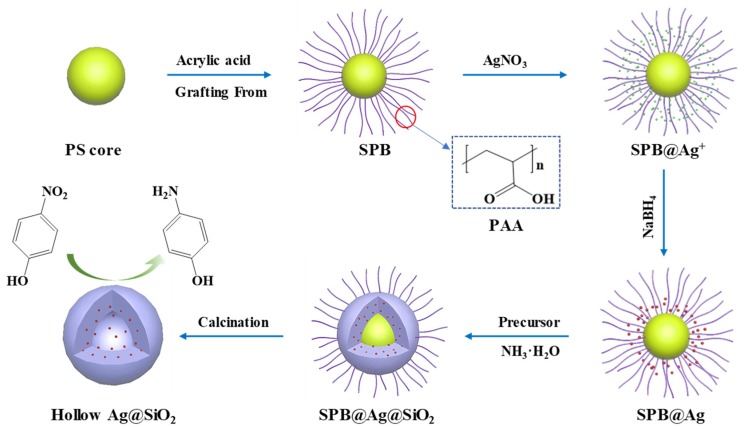
Schematic representation of preparation and catalytic application of hollow silica spheres with encapsulated silver nanoparticles (Ag@SiO_2_) based on SPB.

**Figure 2 nanomaterials-10-00799-f002:**
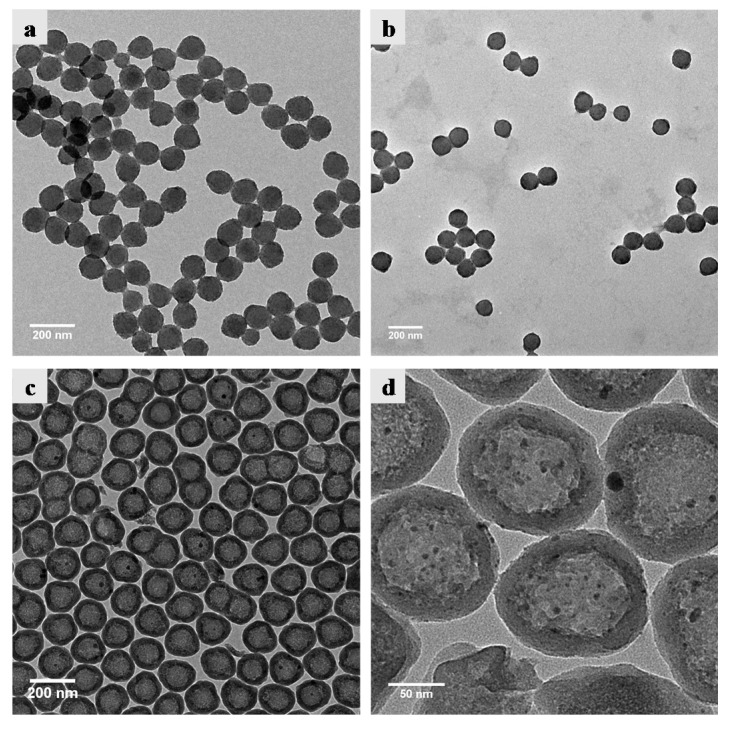
TEM images of (**a**) polystyrene (PS) core, (**b**) spherical polyelectrolyte brushes (SPB), (**c**) hollow Ag@SiO_2_ and (**d**) enlarged hollow Ag@SiO_2_ nanoparticles.

**Figure 3 nanomaterials-10-00799-f003:**
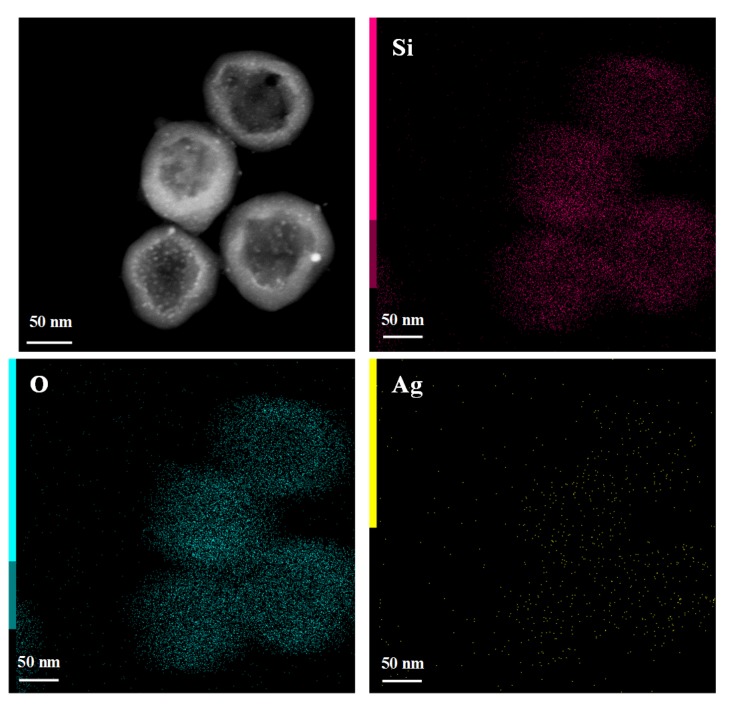
Dark field image of hollow Ag@SiO_2_ spheres and energy-dispersive X-ray spectroscopy (EDX) elemental mapping of Si, O, and Ag elements of hollow Ag@SiO_2_.

**Figure 4 nanomaterials-10-00799-f004:**
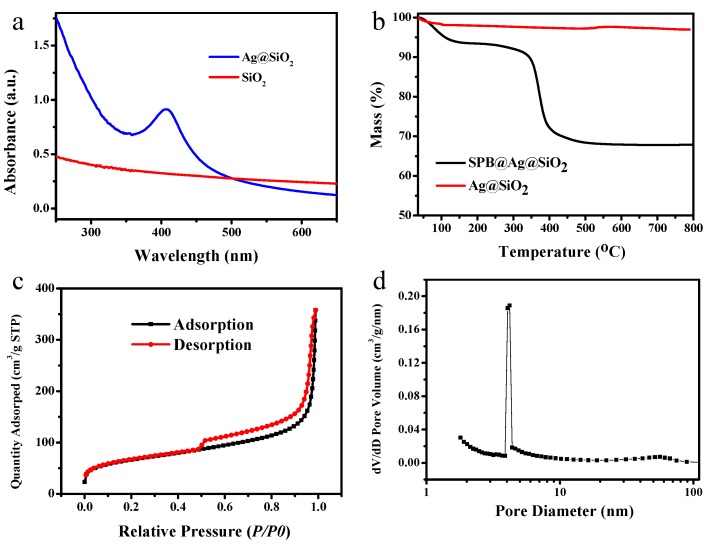
(**a**) UV-vis absorption spectra of hollow SiO_2_ and hollow Ag@SiO_2_; (**b**) TGA curves of SPB@Ag@SiO_2_ and hollow Ag@SiO_2_; (**c**) N_2_ adsorption and desorption isotherms of hollow Ag@SiO_2_ nanoparticles; (**d**) the corresponding pore size distribution calculated by the BJH method.

**Figure 5 nanomaterials-10-00799-f005:**
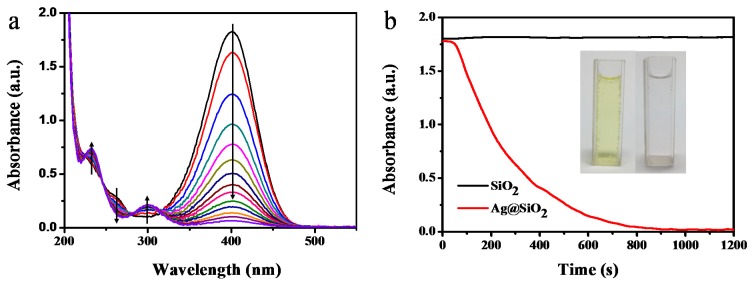
(**a**) The reduction of 4-nitrophenol recorded at different reaction times using hollow Ag@SiO_2_ spheres as the catalyst ([Ag] = 0.27 mg L^−1^ at 298.15 K). The arrows denote the trend of absorbance curve with reaction time (from 0 to 20 min); (**b**) The time dependence of the absorbance of reaction system at 400 nm in the presence of Ag@SiO_2_ and pure SiO_2_ spheres, respectively. The inset picture shows the reaction system with Ag@SiO_2_ as the catalyst before and after reaction.

**Figure 6 nanomaterials-10-00799-f006:**
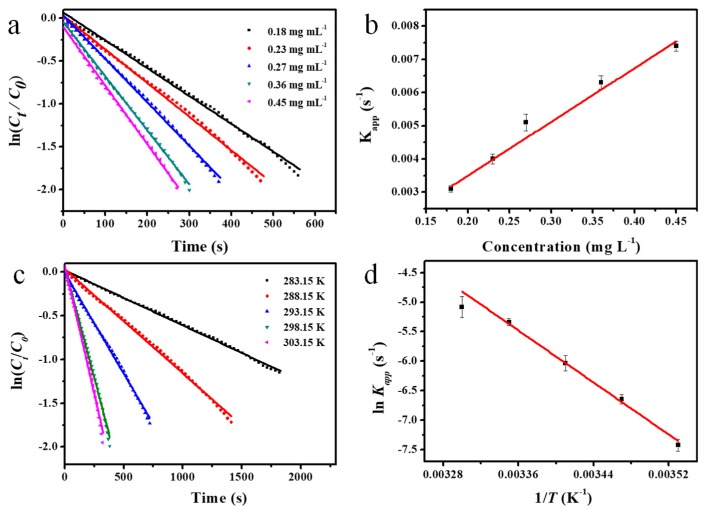
(**a**) Plots of ln (*C_t_/C_0_*) versus reaction time for the reduction of 4-NP catalyzed with different addition of hollow Ag@SiO_2_ at 298.15 K (induction period of the reaction has been subtracted); (**b**) the apparent rate constant (*K_app_*) as a function of the concentration of catalyst; (**c**) plots of ln (*C_t_/C_0_*) versus reaction time for the reduction of 4-NP catalyzed with addition of hollow Ag@SiO_2_ ([Ag] = 0.27 mg L^−1^) at different temperatures (Induction period of the reaction has been subtracted); (**d**) plot of ln *K_app_* versus *1/T*.

**Figure 7 nanomaterials-10-00799-f007:**
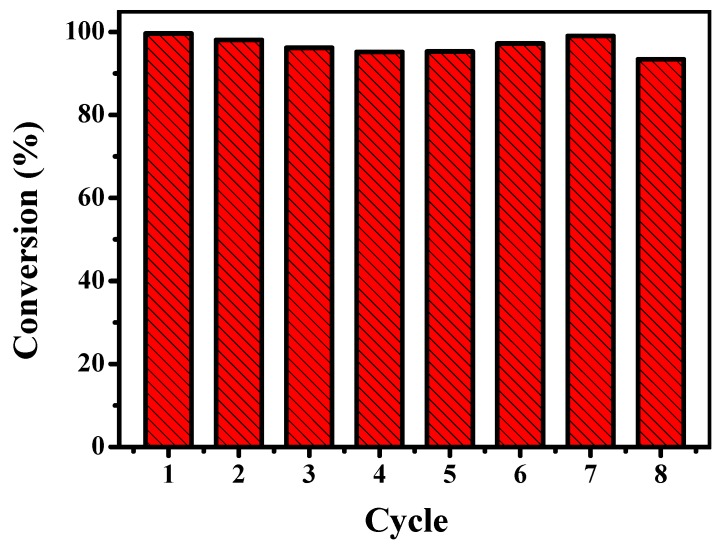
Conversion of the reduction reaction of 4-NP using recollected hollow Ag@SiO_2_ nanospheres as catalyst ([Ag] = 0.27 mg L^−1^, at 298.15 K) for different number of cycles. The Ag@SiO_2_ nanospheres were collected by centrifugation and washed with water twice after each run.

**Figure 8 nanomaterials-10-00799-f008:**
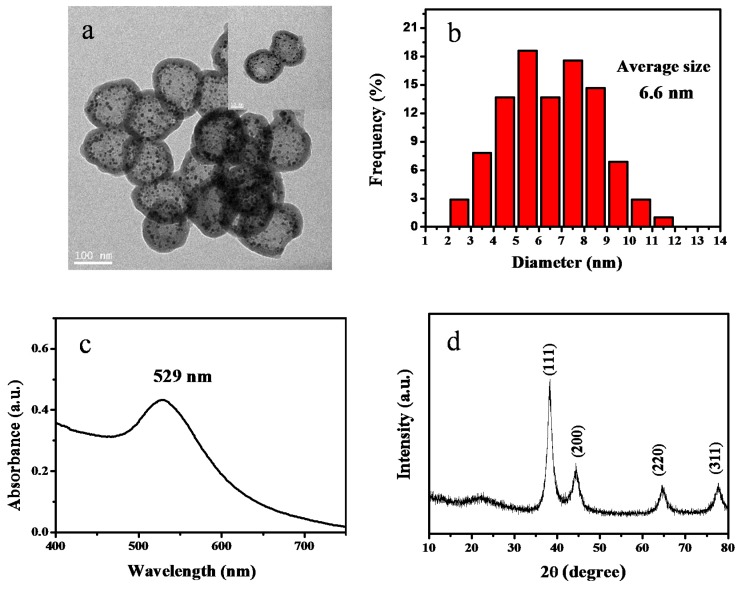
(**a**) TEM images of hollow Au@SiO_2_; (**b**) the corresponding histograms of size distribution of Au nanoparticles; (**c**) UV-vis absorption spectrum, and (**d**) XRD pattern of hollow Au@SiO_2_.

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
