# Peer review of "Spherical Polyelectrolyte Brushes as Templates to Prepare Hollow Silica Spheres Encapsulating Metal Nanoparticles"

_nanomaterials, 2020, doi:10.3390/nano10040799_

Round 1
Reviewer 1 Report
I recommend this manuscript for publication after minor revision, which is desired at following points:
Page 11, line 243:
“.. the silver elements are uniformly distributed in silica shell and the inner space of the hollow sphere.”
Do you mean the inner surface of the hollow sphere?
Page 15, line 289:
“The reaction process can be easily monitored by UV-vis spectroscopy as the 4-NP ions exhibit a typical absorption peak at 400 nm. “
But, Fig. 4 a shows that the typical adsorption peak of silver nanoparticles is also around 400 nm. How was the superposition of the peaks of 4-NP and Ag nanoparticles taken into account when evaluating the measurements?
I miss the Supplementary Materials in the pdf file.
Reviewer 2 Report
This paper builds on earlier work from the same group. The methods are sound, the results are clear. The paper will benefit from a critical last review of the english text. One example: in the abstract, near the end, the word 'metal' was wrongly replaced by 'mental'.
Author Response
We sincerely thank the reviewer’s affirmation of our work and valuable comments. The word “mental” has been changed as “metal”. And we carefully checked the language of the revised manuscript before submission.

Reviewer 3 Report
The submitted manuscript presents the preparation of hollow silica spheres encapsulating silver nanoparticles (Ag@SiO2) or gold nanoparticles (Au@SiO2) based on spherical polyelectrolyte brushes (SPB) which consist of a polystyrene core and densely grafted poly (acrylic acid) (PAA) chains. The resulted hybrid spheres show great potential applications in catalysis.
Some modifications will improve the overall quality of the manuscript. Specific points requiring attention are detailed below.
Comment 1. Line 31 ” mental nanoparticles” should be metal nanoparticles.
Comment 2. Line 39 ….antibacterial [5] should be antibacterial area or antibacterial materials [5].
Comment 3. Add information about city and country of the producer for the delivered materials and for the used apparatuses for characterization.
Comment 4. Display the XDR pattern of Ag@SiO2.
Author Response
Please see the attachment. We highly appreciate the reviewer’s careful reading and valuable comments.
